# Decoding by Contrasting Knowledge: Enhancing LLMs' Confidence on Edited Facts

## Abstract

The knowledge within large language models (LLMs) may become outdated quickly. While in-context editing (ICE) is currently the most effective method for knowledge editing (KE), it is constrained by the black-box modeling of LLMs and thus lacks interpretability. Our work aims to elucidate the superior performance of ICE in KE by analyzing the impacts of in-context new knowledge on token-wise distributions. We observe that despite a significant boost in logits of the new knowledge, the performance of ICE is still hindered by stubborn knowledge. Stubborn knowledge refers to facts that have gained excessive confidence during pretraining, making them hard to edit effectively. To address this issue and further enhance the performance of ICE, we propose a novel approach termed **De**coding by **C**ontrasting **K**nowledge (DeCK). DeCK derives the distribution of the next token by contrasting the logits obtained from the newly edited knowledge guided by ICE with those from the unedited parametric knowledge. Our experiments consistently demonstrate that DeCK enhances the confidence of LLMs in edited facts. For instance, it improves the performance of LLAMA3-8B-INSTRUCT on MQUAKE by up to 219%, demonstrating its capability to strengthen ICE in the editing of stubborn knowledge. DeCK can be easily integrated into any ICE method as a decoding component to enhance editing capabilities. Our work paves the way to develop both effective and accountable KE methods for LLMs.

## 1 Introduction

With the widespread deployment of large language models (LLMs) (OpenAI, 2022; 2023; Touvron et al., 2023a;b; Song et al., 2024), there is a rising demand for accessing accurate information through LLMs. However, despite the extensive knowledge stored in LLMs, this information can become outdated due to changes in the real world. This can potentially result in factual inaccuracies (Chen & Shu, 2023) or false information (Zhang et al., 2023b; Huang et al., 2023a).

Unlike the high computational resource burden incurred by retraining from scratch, knowledge editing (KE) (Sinitsin et al., 2020; De Cao et al., 2021; Zhu et al., 2020; Mitchell et al., 2022; Yao et al., 2023) has been proposed as an efficient means to update the knowledge of LLMs. They aim to edit knowledge by incrementally injecting or modifying facts.

As LLMs demonstrate increasingly powerful in-context learning capabilities, recent research (Madaan et al., 2022; Zhong et al., 2023; Zheng et al., 2023; Cohen et al., 2024; Wang et al., 2024; Bi et al., 2024b;c) has delved into easier and efficient methods for in-context editing (ICE), aiming to directly guide frozen LLMs in generating text with new knowledge through contextual prompts. Figure 1 (left) illustrates an example of successful editing using ICE. These ICE methods showcasing state-of-the-art performance without the need to alter internal model parameters, indicate the promising potential of modeling LLMs as black boxes for ICE guided by external contexts.

However, as illustrated in Figure 1 (middle), there still exist deeply entrenched pieces of knowledge in LLMs that are difficult for ICE to modify, which we refer to as **stubborn knowledge**. We argue that LLMs, through extensive pre-training, have developed strong confidence in certain facts, making them difficult to alter solely through external contextual prompts (Bi et al., 2024a). Therefore, despite the fact that the sophisticated methods such as enhancing retrieval (Shi et al., 2024), checking conflict (Zhong et al., 2023), and guiding reasoning (Wang et al., 2024) can enhance the overall

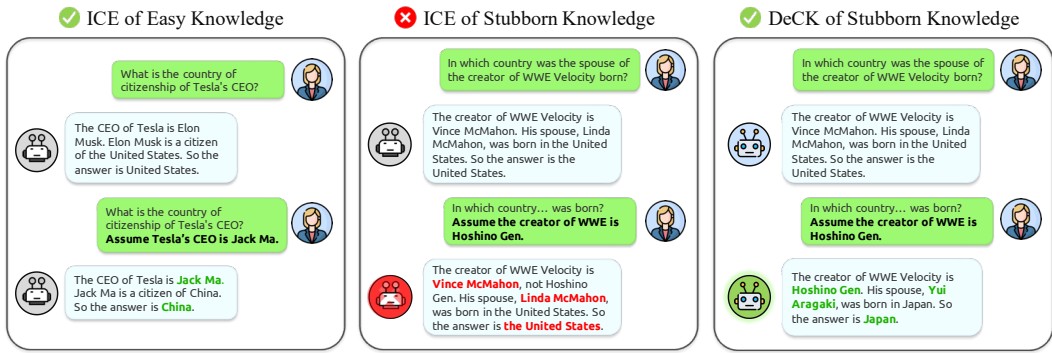

Figure 1: Comparison between in-context editing (ICE) and our DeCK. DeCK successfully edits the stubborn knowledge, whereas ICE handles only simple knowledge and fails with complex cases.

performance of ICE, relying on these external methods still cannot genuinely improve the foundational capability for editing individual stubborn knowledge.

In this work, we focus on enhancing the state-of-the-art KE method, ICE, to reduce the negative impacts from the stubborn knowledge in LLMs. First, we observe the impact of the in-context new knowledge in ICE on LLMs from the perspective of LLMs' token-level distributions. We find that incorporating this new knowledge significantly increases the predicted probability of generating edited facts during the decoding process. A deeper exploration of the failed cases reveals the reasons why stubborn knowledge is difficult to edit. Despite the significant improvement in the logits of new knowledge achieved by ICE, there persists a small gap between new knowledge and parametric knowledge, where parametric knowledge refers to the original unedited knowledge in LLMs.

Building upon the insights gained from above observations, we introduce a new decoding technique called **De**coding by **C**ontrasting **K**nowledge (**DeCK**) to enhance LLMs' confidence in edited facts for better editing of stubborn knowledge. DeCK consists of two components: (1) an editing enhancement module that improves attention to new knowledge, thus preventing it from being filtered out during contrastive decoding, and (2) a contrastive decoding strategy that compares the logical distributions after in-context editing with the original parametric logical distributions to predict the next token.

Overall, our contributions can be summarized by three points. First, as far as we know, we are the first to elucidate superior performance of ICE on the KE from a model interpretability perspective. Second, we find that stubborn knowledge significantly impacts the performance of ICE, and we propose DeCK to boost confidence in editing facts, enhancing ICE to overcome it. Third, extensive experiments on MQUAKE indicate that our DeCK can effectively enhance the performance of ICE without altering the internal model or modifying external prompts. DeCK can be easily integrated into any ICE method as a decoding component to enhance editing capabilities. Our work paves the way to develop the both effective and accountable KE methods for LLMs.

## 2 BACKGROUND

**Decoding in LLMs.** The current objective of LLMs decoding is to predict the subsequent words within a given context sequence. Formally, given a sequence of tokens $\mathcal{X} = \{x_1, x_2, ..., x_{t-1}\}$, the next token probability distribution is computed conditioned on the previous context:

$$\mathbb{P}(x_t|x_{<t}) = \frac{\exp(\mathbf{h}_t^\top \mathbf{W}_{x_t}/\tau)}{\sum_{j \in \mathcal{V}} \exp(\mathbf{h}_t^\top \mathbf{W}_j/\tau)} \quad (1)$$

where $\tau$ represents a temperature parameter regulating the precision of the subsequent-token distribution. In text generation, the language model samples from the conditional distribution $\mathbb{P}(x_t|x_{<t})$ to generate the next token $x_t$, continuing this process until an end-of-sequence token is produced.

**Knowledge Editing.** KE aims to transform the behavior of the original model $f_{base}$ into post-edit model $f_e$. Given an edit descriptor $z_e = (x_e, r_e, y_e)$, where $(x_e, r_e, y_e)$ represents a triplet such as (*US*, *President*, *Joe Biden*) meaning Joe Biden is the president of US. KE ensures that $f_e(x_e, r_e) = y_e$ while $f_{base}(x_e, r_e) \neq y_e$. A thorough edit not only modifies the corresponding

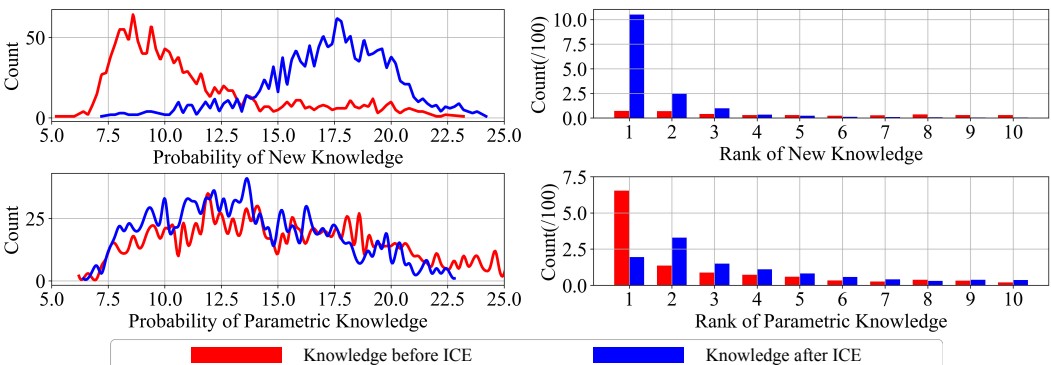

Figure 2: The changes of new knowledge and parametric knowledge before and after editing. We capture the first tokens of outputs to represent the corresponding knowledge and then record their original logits along with their ranks within the entire vocabulary.

knowledge but also all the knowledge within the multi-hop relations that are impacted by this edit. For example, consider a two-hop question like "Who is married to the British Prime Minister?" The original answer would be "Carrie Johnson" and the associated knowledge could be represented: (*UK*, *Prime Minister*, *Boris Johnson*), (*Boris Johnson*, *spouse*, *Carrie Johnson*). With an edit $z_e =$ (*UK*, *Prime Minister*, *Rishi Sunak*) and existing knowledge (*Rishi Sunak*, *spouse*, *Akshata Murthy*), $f_e$ should produce the updated response: "Akshata Murthy".

## 3 DEEP INSIGHTS INTO ICE THROUGH DECODING PERSPECTIVES

With $\phi(\cdot)$ replacing the affine layer to predict the probability of the next token over the vocabulary set $\mathcal{V}$, we can obtain a simplified representation of Equation equation 1. Given a sequence of tokens $\mathcal{X}_E = \{x_1^{(E)}, x_2^{(E)}, ..., x_{m-1}^{(E)}\}$, which includes guidance from an editing prompt, such as "*Assume Tesla's CEO is Jack Ma*", we compute the probability of next token $x_m^{(E)}$ with editing guidance as follows:

$$\mathbb{P}^E(x_m^{(E)}|x_{<m}^{(E)}) = \text{softmax}(\phi(\mathbf{h}_m^{(E)})), \quad x_m^{(E)} \in \mathcal{V} \tag{2}$$

We can also represent the parametric probability distribution $\mathbb{P}^B(x_n^{(B)}|x_{<n}^{(B)})$ by considering only the token sequence $\mathcal{X}_B$ containing the original question prompt without any editing content. The distribution $\mathbb{P}^E(x_m^{(E)}|x_{<m}^{(E)})$ also reflects the feedback from the introduction of external knowledge, while $\mathbb{P}^B(x_n^{(B)}|x_{<n}^{(B)})$ solely represents the response of LLMs based on their parametric knowledge to the question.

### 3.1 HOW ICE CAN EFFECTIVELY EDIT KNOWLEDGE IN LLMS?

Although the ICE methods on LLMs (Zhong et al., 2023; Cohen et al., 2024; Wang et al., 2024) have demonstrated promising performance, they all rely on the black-box modeling of LLMs for editing, and the internal mechanisms behind their effectiveness remain unclear. In this subsection, we delve into the intrinsic reasons behind the superior performance of ICE on KE.

We designed dedicated experiments to capture the logits output of knowledge that would be influenced by the edit. A striking observation in Figure 2 is that the introduction of new knowledge through ICE leads to a significant rightward shift in the probability distribution of the new knowledge, while the logits for parametric knowledge remain largely unchanged or decrease to some extent. This suggests that ICE significantly enhances the logits of new knowledge while having minimal impact on parametric knowledge. Additionally, the number of top-ranked positions for new knowledge significantly increases after ICE, with the majority surpassing that of parametric knowledge. This indicates that the in-context new knowledge can improve the confidence of LLMs in editing facts, thereby prompting responses with the edited answers.

| Case Type | Input | | Knowledge Answer | | Parametric Change | | New Change | |
|---|---|---|---|---|---|---|---|---|
| | question | edit | parametric | new | logits | rank | logits | rank |
| Succ-essful Edit | What's the official language in scr-een International's home country? | The official language of United Kingdom is Italian | Egnlish | Italian | $20.219 \rightarrow 19.875$ | $1 \rightarrow 2$ | $10.461 \rightarrow 20.179$ | $25 \rightarrow 1$ |
| | Which country is the creator of " Devious Maids" a citizen of? | Marc Cherry is a citizen of Bulgaria | Bulgaria | United States | $16.641 \rightarrow 12.211$ | $1 \rightarrow 4$ | $5.586 \rightarrow 18.500$ | $186 \rightarrow 1$ |
| Failed Edit | Which continent does Blur's origin lie in? | London is located in the continent of Australia. | Europe | Austrilia | $27.391 \rightarrow 22.730$ | $1 \rightarrow 1$ | $13.734 \rightarrow 18.094$ | $12 \rightarrow 3$ |
| | What is the official language of the country of Marcellin Champagnat? | The official language of France is English | French | English | $19.266 \rightarrow 17.578$ | $1 \rightarrow 1$ | $12.211 \rightarrow 17.062$ | $4 \rightarrow 2$ |

Figure 3: Edit cases with changes in the first token for both parametric and new knowledge. We obtained the case results by conducting ICE in the LLaMA2-7B-CHAT model. '$\rightarrow$' indicates the knowledge change after incorporating editing prompts. 'logits' and 'rank' pertain to the first token of knowledge answer, reflecting the confidence of LLMs in the corresponding knowledge.

## 3.2 INHERENT CHALLENGES OF STUBBORN KNOWLEDGE

While ICE has significantly boosted the confidence of LLMs in new knowledge, we find that there are still instances where certain new knowledge ranks prominently but not as the top-1, as illustrated in Figure 2. We term this phenomenon "**stubborn knowledge**", which refers to cases where editing fails due to either an excessive confidence in existing parametric knowledge or insufficient confidence in new knowledge. The edit cases in Figure 3 deeply reveals the failed pattern for ICE in addressing stubborn knowledge, which happens when there is still an extremely small gap compared to the parametric knowledge after editing, despite the significant increase in new knowledge logits induced by the editing prompt. Taking the last case as an example, after editing, the new knowledge "English" lags behind the parametric knowledge "French" by only 0.516 in terms of logical distribution, illustrating how a minor gap leads to editing failure. This indicates the intrinsic reasons for the failure of black-box ICE methods to edit stubborn knowledge in LLMs in most cases.

## 4 DeCK: ENHANCING LLMs' CONFIDENCE ON EDITED FACTS

Inspired by the observations in Section 3, we design our novel decoding strategy DeCK to enhance ICE in overcoming stubborn knowledge. Figure 4 illustrates the process of using DeCK to handle the stubborn knowledge case shown in Figure 1 (right). DeCK can be formalized as follows. Using $\mathbb{P}(x_t)$ to represent $\mathbb{P}(x_t|x_{<t})$ for notational brevity, we compute the probability of the next token by,

$$\mathbb{P}_{\text{Enh}}^E(x_m^{(E)}) = \text{Enh}(\mathbb{P}^E(x_m^{(E)})) \tag{3}$$

$$\hat{\mathbb{P}}_{\text{Enh}}^E(x_m^{(E)}) = \text{softmax}\left(\mathcal{F}\left(\mathbb{P}_{\text{Enh}}^E(x_m^{(E)}), \mathbb{P}^B(x_n^{(B)})\right)\right) \tag{4}$$

Here, the function $\text{Enh}(\cdot)$ in Equation 3 is improve the attention to edit facts, as detailed in Section 4.1. The operator $\mathcal{F}(\cdot, \cdot)$ in Equation 4 is used to contrast between the output distributions from enhanced new knowledge and parametric knowledge, as explained in Section 4.2.

### 4.1 EDITING SIGNAL ENHANCEMENT

To enhance the confidence of LLMs in edited knowledge, we design an editing enhancement function that minimizes the Knowledge Enhancement Divergence (KED) between the enhanced distribution and a target distribution. Assume that $\tilde{P}$ and $Q$ are discrete probability distributions over a finite vocabulary $V$, and that the weights $w_i$ are non-negative and sum to 1.

**Definition 4.1** (Knowledge Enhancement Divergence). *Let $\tilde{P}(x_m^{(E)})$ be the enhanced probability distribution of the next token $x_m^{(E)}$ after incorporating edited knowledge, and let $Q$ be the target distribution that assigns higher probabilities to tokens related to the edited knowledge. The KED*

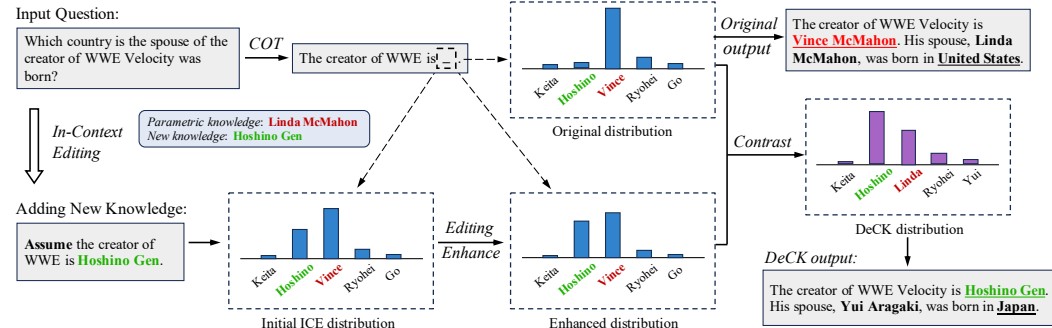

Figure 4: Illustration of DeCK enhancing ICE to edit the stubborn knowledge. During decoding, DeCK contrasts the enhanced ICE distribution with the original distribution to highlight new knowledge, inducing LLMs to generate edited facts using chain-of-though (CoT) (Wei et al., 2022) during the reasoning process for answering input questions.

between $\tilde{P}(x_m^{(E)})$ and $Q$ is defined as:

$$KED(\tilde{P}||Q) = \frac{1}{2} \sum_{i=1}^{n} w_i \left( \tilde{P}(v_i) \log \frac{\tilde{P}(v_i)}{M(v_i)} + Q(v_i) \log \frac{Q(v_i)}{M(v_i)} \right) \tag{5}$$

where $M = \frac{1}{2}(\tilde{P} + Q)$ is the average distribution, and $w_i = s(v_i, E)$ is the weight assigned to the $i$-th token based on its semantic relevance score.

We introduce a semantic relevance function $s : V \times E \to \mathbb{R}$ that measures the relevance of a token $v_i \in V$ to the edited knowledge represented by $E$, defined as:

$$s(v_i, E) = \max_{e_j \in E} \text{sim}(v_i, e_j) \cdot \phi(v_i)$$

where $\text{sim}(\cdot, \cdot)$ is a similarity function, such as cosine similarity, that measures the semantic similarity between two token embeddings, and $\phi : V \to \mathbb{R}$ is a frequency-based weighting function:

$$\phi(v_i) = \log(\text{freq}(v_i) + \epsilon) \cdot \alpha$$

Here, freq $: V \to \mathbb{N}$ denotes the frequency of a token in the edited descriptor $E$, $\epsilon > 0$ is a small constant to avoid taking the logarithm of zero, and $\alpha$ is a scaling factor. We also define an enhancement function Enh $: \mathbb{R}^n \times \mathbb{R}^n \to \mathbb{R}^n$ that takes the original logits $\phi(h_m^{(E)}) \in \mathbb{R}^n$ and the semantic relevance scores $s \in \mathbb{R}^n$ as inputs and produces the enhanced logits $\tilde{\phi}(h_m^{(E)}) \in \mathbb{R}^n$:

$$\text{Enh}(\phi(h_m^{(E)}), s) = \alpha \cdot \phi(h_m^{(E)}) + \beta \cdot s$$

where $\alpha, \beta \in \mathbb{R}$ are scaling coefficients that control the balance between the original logits and the semantic relevance scores. Hence, the target distribution $Q$ over the vocabulary $V$ is constructed to assign higher probabilities to the tokens related to the edited knowledge:

$$Q(v_i) = \begin{cases} \frac{1}{m} & \text{if } v_i \in E \\ \epsilon & \text{otherwise} \end{cases}$$

where $\epsilon > 0$ is a small constant to ensure a valid probability distribution.

## 4.2 DECODING BY CONTRASTING KNOWLEDGE

The idea of Decoding by Contrasting Knowledge is to highlight the output probability increment of new knowledge by contrasting it with the parametric knowledge from the inherent knowledge of the LLMs. Given the ICE probability distribution $\mathbb{P}_{\text{Enh}}^E(x_m^{(E)})$ after editing enhancement in Section 4.1 and the original parametric probability distribution $\mathbb{P}^B(x_n^{(B)})$, we aim to amplify the outputs of new knowledge during the generation process while downplaying the outputs of paprametric knowledge.

| Model | Method | MQUAKE-3K | MQUAKE-2002 | MQUAKE-HARD |
|---|---|---|---|---|
| LLAMA2-7B-CHAT | ROME (Meng et al., 2022a) | 18.2 | 19.1 | 15.7 |
| | IKE (Zheng et al., 2023) | 85.4 | 85.1 | 88.9 |
| | IKE w/ DeCK (ours) | **91.3** | **89.4** | **98.6** |
| LLAMA2-13B-CHAT | ROME (Meng et al., 2022a) | 39.4 | 39.7 | 35.2 |
| | IKE (Zheng et al., 2023) | 63.8 | 64.1 | 55.2 |
| | IKE w/ DeCK (ours) | **84.6** | **84.4** | **89.7** |
| LLAMA3-8B-INSTRUCT | ROME (Meng et al., 2022a) | 14.5 | 15.9 | 12.7 |
| | IKE (Zheng et al., 2023) | 31.6 | 32.5 | 14.3 |
| | IKE w/ DeCK (ours) | **54.7** | **55.9** | **45.7** |
| MISTRAL-7B-INSTRUCT | ROME (Meng et al., 2022a) | 28.1 | 30.2 | **26.3** |
| | IKE (Zheng et al., 2023) | 34.1 | 35.6 | 15.6 |
| | IKE w/ DeCK (ours) | **46.7** | **48.5** | 19.2 |

Table 1: Experimental results (accuracy; %) across various models and datasets. We set the batch size of the edit memory to 1 to evaluate the foundational capability of directly editing knowledge. The best editing result on every LLM is highlighted in bold font.

Following the Contrastive Decoding approach proposed by Li et al. (2023). We subtract the original log probabilities of parametric outputs guided by knowledge question alone from those of the outputs guided by ICE with the in-context new knowledge. Then, we use this resulting distribution as the next-word prediction for the generation guided by editing prompts. Therefore, the operator $\mathcal{F}(\cdot, \cdot)$ in Equation 4 can be expanded as follows:

$$
\mathcal{F}(\mathbb{P}_{\text{Enh}}^E(x_m^{(E)}), \mathbb{P}^B(x_n^{(B)})) = \begin{cases} \log \dfrac{\mathbb{P}_{\text{Enh}}^E(x_m^{(E)})}{\mathbb{P}^B(x_n^{(B)})}, & \text{if } x_m^{(E)} \in \mathcal{V}_{\text{head}}\left(x_m^{(E)} | x_{<m}^{(E)}\right), \\ -\infty, & \text{otherwise.} \end{cases} \tag{6}
$$

The contrasting coefficient $\gamma$ is also introduced to adjust the proportion of the subtraction: $\log \mathbb{P}_{\text{Enh}}^E(x_m^{(E)}) - \gamma \log \mathbb{P}^B(x_n^{(B)})$. And the subset $\mathcal{V}_{\text{head}}\left(x_m^{(E)} | x_{<m}^{(E)}\right) \in \mathcal{V}$ is defined as whether or not the token has high enough probabilities from the editing output,

$$
\mathcal{V}_{\text{head}}(x_m^{(E)} | x_{<m}^{(E)}) = \left\{ x_m^{(E)} \in \mathcal{V} : \mathbb{P}_{\text{Enh}}^E(x_m^{(E)}) \geq \lambda \max_w \mathbb{P}_{\text{Enh}}^E(w) \right\}. \tag{7}
$$

As adaptive plausibility constraint (APC) strategy proposed in Li et al. (2023), we use $\mathcal{V}_{\text{head}}$ to filter out tokens with low probabilities in $\mathbb{P}_{\text{Enh}}^E(x_m^{(E)})$ and considering only high-score tokens. Without APC, some extremely low-probability tokens could be excessively amplified by the softmax function after subtraction, leading to the generation of implausible words and severely impacting the performance of contrastive decoding. Specifically, the Editing Signal Enhancement module in Section 4.1 cleverly avoids being filtered out in Equation 7 by enhancing the new knowledge signal before contrastive processing, ensuring that DeCK can function effectively.

The key to our contrastive decoding approach is the simultaneous maintenance of two token sequences' generation, which differs from previous methods (Li et al., 2023; Chuang et al., 2023; Zhang et al., 2023a). In iterative decoding, we predict the next token based on $\hat{\mathbb{P}}_{\text{Enh}}^E(x_m^{(E)})$ in Equation 4. Then, a key step involves simultaneously concatenating the new token to two separate token sequences $\mathcal{X}_E$ and $\mathcal{X}_B$, which may have different lengths. This ensures that updates to both sequences are synchronized, preventing any implausible discrepancies in the log distribution during iteration.

## 5 EXPERIMENTS

### 5.1 EXPERIMENTAL SETUP

**Datasets** We conduct extensive experiments using the MQUAKE-3K dataset (Zhong et al., 2023) and its derivatives, MQUAKE-2002 and MQUAKE-HARD, proposed by Wang et al. (2024).

| Model | Method | MQUAKE-3K | MQUAKE-2002 | MQUAKE-HARD |
|-------|--------|-----------|-------------|-------------|
| LLAMA2-7B-CHAT | IKE (Zheng et al., 2023) | 20.7 | **20.6** | 2.3 |
| | IKE w/ DeCK (ours) | **22.4** | 20.4 | **3.8** |
| | MeLLo (Zhong et al., 2023) | 32.6 | 40.8 | 5.1 |
| | MeLLo w/ DeCK (ours) | **43.1** | **45.8** | **5.8** |
| LLAMA2-13B-CHAT | IKE (Zheng et al., 2023) | 19.4 | **18.8** | 2.7 |
| | IKE w/ DeCK (ours) | **20.6** | 18.4 | **3.5** |
| | MeLLo (Zhong et al., 2023) | 33.4 | 35.9 | 3.9 |
| | MeLLo w/ DeCK (ours) | **36.8** | **38.2** | **6.2** |

Table 2: Experimental results (accuracy; %) using LLAMA2-CHAT models. We conduct the experiments with the full batch size edit memory to evaluate the performance of memory based KE.

MQUAKE provides multi-hop knowledge questions containing extensively edited facts, which are used to evaluate KE on counterfactual edits. Additionally, we constructed corresponding STUBBORN datasets in 5.3 to further evaluate the effectiveness of editing stubborn knowledge.

**Models and Baselines**  Our experiments examine three types of LLAMA-CHAT models (2-7b, 2-13b, 3-8b) (Touvron et al., 2023b) and also MISTRAL-7B-INSTRUCT (Jiang et al., 2023). We employ the state-of-art in-context editing methods IKE (Cohen et al., 2024) and MeLLo (Zhong et al., 2023), alongside advanced model-editing techniques ROME (Meng et al., 2022a) as baseline approaches on the aforementioned open-source models. IKE prompts LLMs to edit given knowledge by providing contextual demonstrations. MeLLo edits multi-hop knowledge by decomposing sub-questions, prompting LLMs to generate answers, and retrieving contradictions from the edit memory.

**Implementation Details**  We implement IKE with multi-hop question-answering demonstrations and chain-of-thought (COT) (Wei et al., 2022; Li et al., 2024) prompting to enhance its in-context editing performance. Our decoding strategy DeCK is inherently deployed onto IKE and MeLLo to validate their enhancements without any additional adjustments. It simply requires providing the relevant factual guiding context before generating the edited answers. The model editing methods ROME in our baselines are deployed using EasyEdit (Wang et al., 2023). We set adaptive plausibility constraint $\lambda$ to 0.01 and contrasting coefficient $\gamma$ to 0.2 for our DeCK.

## 5.2 MAIN RESULTS

We evaluate the foundational capability of KE methods in directly editing explicit new knowledge by considering multi-hop questions containing 1,000 instances and setting the batch size of the edit memory to 1. The batch size means the number of instances providing the edited facts for knowledge retrieval. Table 1 displays the performance of different baselines and the enhanced in-context editing through our DeCK across various models and datasets. As with previous work, ICE methods exhibit superior performance in multi-hop KE tasks compared to model-editing methods ROME. Overall, IKE enhanced by our DeCK (IKE w/ DeCK) consistently exhibits the best performance, indicating that the DeCK can reliably improves the foundational KE capabilities of ICE for LLMs. Specifically, as the model parameters increase, LLMs tend to retain more stubborn knowledge, resulting in a decrease in the accuracy of ICE. For instance, the average accuracy of LLAMA2-13B-CHAT is 61%, whereas that of LLAMA2-7B-CHAT is 86%. Additionally, although the parameters of llama3 are not extensive, its more refined pretraining and instruct tuning also may instill greater confidence in its acquired knowledge, resulting in poor performance in ICE. However, to our great surprise, our DeCK has significantly enhanced ICE's editing of these stubborn knowledge. Notably, on the HARD dataset, DeCK has increased ICE's editing success rate in LLAMA2-13B-CHAT by an impressive 63% and in LLAMA3-8B-INSTRUCT by an amazing 219%.

In-context editing methods typically require retrieving edit demonstrations from the edit memory and then editing LLMs with the retrieved knowledge. Therefore, we follow the setup of previous work (Zheng et al., 2023; Zhong et al., 2023; Madaan et al., 2022) to conduct experiments for ICE methods with the full batch size edit memory. As shown in Table 2, the experimental results illustrate that DeCK enhances ICE methods to varying degrees in full batch experiments. The IKE methods does not

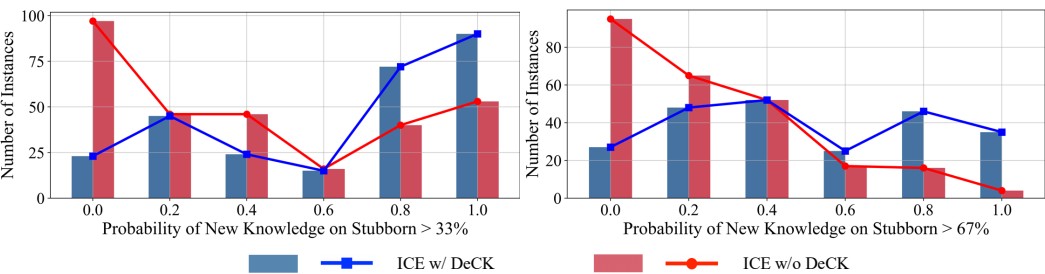

Figure 5: Probability statistics of new Knowledge for LLAMA2-7B-CHAT on stubborn datasets. The probabilities are derived from softmax calculations.

exhibit consistent improvement in this regard, potentially constrained by its inherent editing accuracy. We ingeniously integrate our DeCK into MeLLo, aiding MeLLo in generating crucial edited answers during the reasoning process. We find that leveraging the foundational editing capabilities provided by DeCK consistently improves MeLLo's performance across all experiments. This indicates that our DeCK holds significant potential for real-world KE applications.

| Original Rank | 2 | 3-5 | 6-10 | 11-20 | 21-50 | 51-100 |
|---|---|---|---|---|---|---|
| LLAMA2-7B-CHAT | $1.6_{(\uparrow 0.4)}$ | $2.7_{(\uparrow 0.9)}$ | $4.3_{(\uparrow 3.6)}$ | $4.6_{(\uparrow 8.2)}$ | $4.8_{(\uparrow 24.3)}$ | $6.1_{(\uparrow 61.3)}$ |
| LLAMA2-13B-CHAT | $1.4_{(\uparrow 0.6)}$ | $1.9_{(\uparrow 1.9)}$ | $2.2_{(\uparrow 4.9)}$ | $2.8_{(\uparrow 13.4)}$ | $4.1_{(\uparrow 34.1)}$ | $5.4_{(\uparrow 72.7)}$ |

Table 3: Improvement of new knowledge ranking by DeCK on MQUAKE-3K. Here, 'original rank' refers to the ranking of new knowledge after the original IKE w/o DeCK. The table shows the average ranking of new knowledge and the improvement after integrating DeCK.

## 5.3 METAMORPHOSIS OF STUBBORN KNOWLEDGE

To further explore the reasons behind the significant improvement brought by DeCK to ICE, we conduct a statistical analysis of the ranking changes. Specifically, we sample the new knowledge with probability rankings between top 2-100 after the original ICE method, and examine the changes in their ranks after integrating DeCK. The results in Table 3 demonstrate that our DeCK effectively improves the ranking of new knowledge that failed to be edited by ICE, leading to a metamorphosis of stubborn knowledge.

| Model | STUBBORN | ROME | IKE | IKE w/ DeCK (ours) |
|---|---|---|---|---|
| LLAMA2-7B-CHAT | $> 33\%$ | 17.7 | 56.4 | **72.3** |
| | $> 67\%$ | 19.3 | 37.8 | **55.9** |
| LLAMA2-13B-CHAT | $> 33\%$ | 42.5 | 38.9 | **70.1** |
| | $> 67\%$ | 40.2 | 29.4 | **48.5** |

Table 4: Performance of LLAMA2-7B-CHAT and LLAMA2-13B-CHAT on their respective STUB-BORN datasets. 'STUBBORN > 67%' indicates instances from the MQUAKE-3K dataset where IKE failed to edit knowledge more than 67% of the time. 'STUBBORN > 33%' follows the same criterion.

We constructed corresponding STUBBORN datasets for different models to specifically evaluate ICE's performance on stubborn knowledge. The stubborn datasets are categorized into different difficulty levels based on the proportion of correct answers when using ICE methods to edit the same knowledge multiple times with different knowledge questions. The experimental results on the STUBBORN datasets are presented in Table 4. We found that IKE's performance on STUBBORN significantly declined compared to other datasets, as shown in Table 1, and even fell below that of the model editing method ROME on LLAMA2-13B-CHAT. Our DeCK consistently brings about a dramatic improvement for IKE, with enhancements of up to 80% on LLAMA2-13B-CHAT, ensuring that IKE w/ DeCK maintains the highest performance. This suggests that DeCK brings about improvements by enhancing ICE methods' ability to edit stubborn knowledge.

Figure 5 reveals the underlying reasons why DeCK can effectively edit stubborn knowledge. ICE w/ DeCK has a higher distribution in the high-probability range, while ICE w/o DeCK is concentrated

in the low-probability range. This further indicates that DeCK boosts the confidence of LLMs in low-confidence new knowledge, making them more likely to accept the edited facts.

## 5.4 ABLATION STUDY

We conduct ablation experiments on the key components of our DeCK. Table 5 shows how the contrasting coefficient introduced in Equation 6 affects DeCK's performance. DeCK is highly sensitive to the contrasting coefficient. If $\gamma$ is too large, it can excessively amplify unreasonable token probabilities, significantly reducing DeCK's performance, even below that of the original ICE. Table 6 demonstrates that the editing signal enhancement introduced in Section 4.1 can consistently enhance DeCK's performance. This is because it ensures that the enhanced edited knowledge is not filtered out by Equation 7.

| Scale | $\gamma = 0.1$ | $\gamma = 0.2$ | $\gamma = 0.5$ |
|-------|------|------|------|
| -7B | 88.7 | **91.3** | 80.2 |
| -13B | 76.1 | **84.6** | 48.5 |

| DeCK | MQUAKE-3K | MQUAKE-2002 |
|------|------|------|
| w/o Enh | 89.1 | 87.3 |
| w/ Enh | **91.3** | **89.4** |

Table 5: Ablation results on MQUAKE-3K with LLAMA2-CHAT models.

Table 6: Ablation results of the editing signal enhancement component on the LLAMA2-7B-CHAT model.

## 6 RELATED WORK

**Hallucinations and Misinformation**   Hallucination (Kang et al., 2024) is one of the main source of LLM-generated misinformation. In general, there are two lines of works on hallucination mitigation. In training stage, Hu et al. (2023); Pan et al. (2024) has investigated training data curation or knowledge grounding methods to integrate more knowledge. In the inference stage, recent works have explored methods including confidence estimation (Huang et al., 2023b), knowledge retrieval (Feng et al., 2024; Yang et al., 2024) and KE to improve accurate outputs.

**Contrast Decoding**   The recent contrasting decoding methods achieve the desired output by contrasting logical distribution during the decoding phase. CD (Li et al., 2023) compares powerful expert language models with weaker amateur language models to enhance fluency and coherence. DoLa (Chuang et al., 2023) contrasts mature layers with premature layers, while ICD (Zhang et al., 2023a) compares with models injected with hallucinations, aiming to enhance the factual accuracy of the model.

**Model Editing and In-Context Editing**   Model Editing is a type of effective technique for KE, altering the model's internal structure to modify its output regarding the edited content. Current model editing methods (Meng et al., 2022a;b; Mitchell et al., 2022; Yao et al., 2023; Xu et al., 2024) for LLMs involve integrating an auxiliary network with the original model or modifying and adding model parameters to manipulate the model's output. The emergent method of ICE (Madaan et al., 2022; Zhong et al., 2023; Zheng et al., 2023), demonstrates significant potential, enabling the editing of language models by prompting them with edited fact and retrieving editing demonstrations from the edit memory.

## 7 CONCLUSION AND LIMTATIONS

In this paper, we introduce Decoding by Contrasting Knowledge (DeCK), a novel decoding strategy aimed at enhancing in-context editing in overcoming stubborn knowledge for LLMs. Based on observations at the token-level of edited knowledge, DeCK contrasts the logits of new knowledge with those from parametric knowledge to amplify the changes in model knowledge brought about by in-context editing. Experimental results show that DeCK significantly improves editing accuracy. Overall, DeCK is a critical step in enhancing in-context editing to overcome stubborn knowledge.

DeCK also has limitations; it requires the reception of input from two different token sequences during the generation process, resulting in approximately a 1.6X increase in latency compared to original decoding. This suggests that we can pursue further optimization within the transformers architecture or explore alternative, more cost-effective versions of DeCK.

ETHICS STATEMENT

Ethical considerations are of utmost importance in our research endeavors. In this paper, we conscientiously adhere to ethical principles by exclusively utilizing open-source datasets and employing models that are either open-source or widely recognized in the community. Moreover, our proposed method is designed to ensure that the model does not produce any harmful or misleading information. We are committed to upholding ethical standards throughout the research process, prioritizing transparency, and promoting the responsible use of technology for the betterment of society.

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
