# OpenReview forum: "Decoding by Contrasting Knowledge: Enhancing LLMs' Confidence on Edited Facts"
_ICLR.cc/2025/Conference — ICLR 2025 Conference Withdrawn Submission_

### Official Review · Reviewer_mLwR · 2024-10-29

**Soundness:** 2
**Presentation:** 2
**Contribution:** 2
**Rating:** 5
**Confidence:** 4

**Summary:**

This paper examines the performance of In-Context Editing (ICE) for knowledge editing in large language models (LLMs) and introduces "Decoding by Contrasting Knowledge" (DeCK) to enhance ICE’s effectiveness. DeCK achieves this by contrasting the logits of newly edited knowledge with the model's original parametric knowledge, thereby strengthening the model's adherence to updated information. Experimental results on the MQUAKE dataset demonstrate performance improvements, particularly in handling difficult-to-edit "stubborn knowledge".

**Strengths:**

1. The paper offers insights into in-context editing (ICE) by exploring it through a decoding lens, enhancing the understanding of ICE's mechanisms and its limitations in knowledge editing.

2. Extensive experiments showcase the effectiveness of DeCK in tackling stubborn knowledge, demonstrating significant performance improvements and validating its impact across challenging cases.

3. The paper is well presented.

**Weaknesses:**

1. The concept of "stubborn knowledge" is introduced but lacks a precise, quantifiable definition. A clearer, potentially quantitative description of what constitutes stubborn knowledge would help clarify how DeCK differentiates from conventional methods.

2. The paper would benefit from a deeper examination of why current IKE methods fall short in handling certain cases, particularly where stubborn knowledge resists alteration. A more detailed analysis of these limitations and how DeCK specifically overcomes them would enhance the understanding of DeCK’s improvements.

3. DeCK requires generating inferences twice for each instance, increasing computational demands. This added cost may be a barrier for practical applications.

**Questions:**

1. How does model size impact DeCK's performance? For instance, would DeCK show different results if applied to much larger models, such as the LLAMA 3 70B?

2. How would DeCK perform on datasets beyond MQUAKE? The experiments primarily use the MQUAKE dataset and its derivatives; would DeCK’s effectiveness hold across other types of datasets?

---

### Official Review · Reviewer_eNre · 2024-10-31

**Soundness:** 2
**Presentation:** 2
**Contribution:** 2
**Rating:** 3
**Confidence:** 4

**Summary:**

To improve the performance of in-context editing methods on multi-hop knowledge editing tasks, this paper introduces a new decoding method DeCK to lead the language model output the new knowledge. In detail, DeCK uses an editing enhancement module that improves attention to new knowledge and uses a contrastive decoding strategy to predict the new knowledge. The experiments show that the in-context editing with DeCK outperforms the in-context editing methods a lot.

**Strengths:**

1. This paper finds that previous in-context editing methods fail to follow new knowledge and propose new decoding methods to address the weakness.

2. This paper conducts a detailed ablation study to verify the effectiveness of the proposed method.

**Weaknesses:**

The main weakness of this method is the absence of robust baselines for fair comparison. While ROME and MELO serve as overly simplistic baselines for multi-hop editing tasks in MQUAKE, there are several other methods, such as RAE [1] and Pokemqa [2], that have gained acceptance.

2. DeCK specifically targets single-fact editing, as in-context editing simplifies complex multi-hop questions into single-hop ones. Therefore, DeCK should be compared with traditional knowledge editing tasks. For instance, IKE [3] closely aligns with DeCK’s objectives, using external example prompts to enable LLMs to incorporate new knowledge effectively.
[1] Shi, Yucheng, et al. "Retrieval-enhanced knowledge editing for multi-hop question answering in language models." arXiv preprint arXiv:2403.19631 (2024).
[2] Gu, Hengrui, et al. "Pokemqa: Programmable knowledge editing for multi-hop question answering." arXiv preprint arXiv:2312.15194 (2023).
[3] Zheng, Ce, et al. "Can we edit factual knowledge by in-context learning?" arXiv preprint arXiv:2305.12740 (2023).

**Questions:**

Will you consider to add baselines?

---

### Official Review · Reviewer_2BuY · 2024-11-01

**Soundness:** 2
**Presentation:** 3
**Contribution:** 2
**Rating:** 3
**Confidence:** 4

**Summary:**

The authors suggest that stubborn knowledge hinders the performance of in-context editing.
Then, the authors introduce Decoding by Contrasting Knowledge (DeCK), a novel decoding strategy aimed at enhancing in-context editing in overcoming stubborn knowledge for LLMs.
Experimental results show that DeCK significantly improves editing accuracy.

**Strengths:**

The writing in this article is well-organized, and the proposed approach is clear and easy to understand.

**Weaknesses:**

1. The experiment is insufficiently thorough: The baseline only includes naive versions of ROME and IKE, overlooking other types of knowledge editing methods such as SERAC and MEND.

2. The evaluation metrics for the results seem to overlook several common editing metrics, such as locality.

3. The proposed method aims to enhance the model’s reliance on in-content text rather than on stubborn parametric knowledge. Intuitively, this would likely increase the chances of incorporating new knowledge in responses. However, it also **introduces safety risk** by raising the likelihood of the model being bypassed by jailbreak prompts. Specifically, if malicious knowledge is appended to the query, this approach heightens the probability of the model using that malicious information.

**Questions:**

Has the author considered the potential security risks introduced by the proposed method?

**Details Of Ethics Concerns:**

The method proposed by this paper may introduce safety risk by raising the likelihood of the model being bypassed by jailbreak prompts. Specifically, if malicious knowledge is appended to the query, this approach heightens the probability of the model using that malicious information.

---

> ### Comment · Reviewer_2BuY · 2024-11-22
> **Some additional supplements to the feedback provided by the Associate Program Chairs above.**
>
> Thanks to the Associate Program Chairs for their suggestions.
>
> 1. The purpose of editing is to correct inaccurate knowledge while minimizing the side effects caused by the editing process. In other words, unrelated knowledge should remain unaffected, which is measured by the locality [1] metric.
>
> 2. Since MEND and SERAC are representative editing methods that have achieved good performance across various metrics, it is recommended that the authors evaluate MEND and SERAC against DeCK.
>
> 3. The proposed method aims to enhance the model’s reliance on in-content text rather than on stubborn parametric knowledge. Intuitively, this would likely increase the chances of incorporating new knowledge in responses. However, it also introduces safety risk by raising the likelihood of the model being bypassed by jailbreak prompts. Specifically, if malicious knowledge is appended to the query, this approach heightens the probability of the model using that malicious information.
>
> - Therefore, it is suggested that the authors include additional experiments evaluating the method's vulnerability to malicious inputs, in order to minimize potential security risks.
> - Additionally, has the author considered these security risks and designed specific mitigation strategies they might implement to address these concerns?
> Reference
>
> [1] Editing Large Language Models: Problems, Methods, and Opportunities, EMNLP 2023

---

### Official Review · Reviewer_akNa · 2024-11-04

**Soundness:** 3
**Presentation:** 3
**Contribution:** 2
**Rating:** 5
**Confidence:** 4

**Summary:**

This paper introduces Decoding by Contrasting Knowledge (DeCK), a decoding strategy for editing knowledge in LLMs -- DeCK addressses the problem that, even when using in-contextk nowledge editing, the model might still decide to rely on its parametric knowledge (this phenomenon is referred to as "stubborn knowledge" in this paper). DeCK relies on contrastive decoding: during generation, it decreases the likelihood of the tokens that would have been selected without in-context editing, and increases the likelihood of the tokens that would be selected with in-context editing. Contrastive decoding (CD) is an extremely effective technique that has been used in many uses cases recetly, such as mitigating faithfulness hallucinations (e.g., https://arxiv.org/abs/2305.14739, https://arxiv.org/abs/2410.18860, https://arxiv.org/abs/2309.03883), alignment (e.g., https://arxiv.org/abs/2401.08565), and controlled generation (https://aclanthology.org/2021.acl-long.522/) -- this paper is yet another success story of CD.

To assess the effectiveness of DeCK, the authors use the MQuaKE dataset -- why? There are several widely used knowledge conflicts datasets in the literature, like NQ-Swap (https://arxiv.org/abs/2109.05052), MacNoise (https://aclanthology.org/2024.findings-naacl.159/), and ConflictBank (https://arxiv.org/pdf/2408.12076); why did they go specifically for MQuaKE, and only considered MQuaKE variants? Does the proposed approach generalise beyond MQuaKE?

The authors say that "we are the first to elucidate superior performance of ICE on the KE" -- however, the shortcomings of other knowledge editing approaches compared to in-context editing were already shown by e.g. https://arxiv.org/abs/2306.09306

The authors consider a very tiny number of knowledge editing baselines -- mainly ROME, IKE, and MeLLo. However, there is a large number of knowledge editing approaches, including ones that aim to bridge the gap between parameter editing and in-context editing (e.g., https://arxiv.org/abs/2306.09306); even ROME has a set of more efficient and effective developments (e.g., MEMIT: https://memit.baulab.info/). Some potential approaches the authors could experiment with for increasing the reliance of the model on the edited knowledge are listed in e.g., https://arxiv.org/abs/2410.15999 Tab. 1 (DoLA, SAE, CAD, etc.). How does DiKE compare to this? Note that many of these rely on contrastive decoding and/or do not require any additional gradient-based training, so it should be easy to include these in the comparison.

**Strengths:**

Efficient and simple method for in-context editing of knowledge in LLMs that does not require access to the model parameters.

**Weaknesses:**

Lack of comparison baselines. For example, the proposed method is very similar to CAD (https://arxiv.org/abs/2305.14739), that aims to do something very similar (increase reliance on a given context), but there is no comparison with CAD and related methods.
I mentioned other potential weaknesses in the Summary (e.g., lack of datasets), but this one might be the main weakness.

**Questions:**

How does the proposed method compare to e.g., CAD, and other approaches in this space?
Does the method generalise beyond MQuaKE? What about NQ-Swap?

---

### Note · Authors · 2024-11-24

I have read and agree with the venue's withdrawal policy on behalf of myself and my co-authors.